# Mitochondrial Damage-Associated Molecular Patterns Content in Extracellular Vesicles Promotes Early Inflammation in Neurodegenerative Disorders

**DOI:** 10.3390/cells11152364

**Published:** 2022-08-01

**Authors:** Cláudia M. Deus, Henrique Tavares, Margarida Beatriz, Sandra Mota, Carla Lopes

**Affiliations:** CNC—Center for Neuroscience and Cell Biology, CIBB—Center for Innovative Biomedicine and Biotechnology, III-Institute of Interdisciplinary Research, University of Coimbra, 3030-789 Coimbra, Portugal; cmcdeus@cnc.uc.pt (C.M.D.); henriquer99@hotmail.com (H.T.); margaridabeatriz@live.com.pt (M.B.)

**Keywords:** extracellular vesicles, neurodegenerative disorders, mitochondrial damage-associated molecular patterns, inflammation

## Abstract

Neuroinflammation is a common hallmark in different neurodegenerative conditions that share neuronal dysfunction and a progressive loss of a selectively vulnerable brain cell population. Alongside ageing and genetics, inflammation, oxidative stress and mitochondrial dysfunction are considered key risk factors. Microglia are considered immune sentinels of the central nervous system capable of initiating an innate and adaptive immune response. Nevertheless, the pathological mechanisms underlying the initiation and spread of inflammation in the brain are still poorly described. Recently, a new mechanism of intercellular signalling mediated by small extracellular vesicles (EVs) has been identified. EVs are nanosized particles (30–150 nm) with a bilipid membrane that carries cell-specific bioactive cargos that participate in physiological or pathological processes. Damage-associated molecular patterns (DAMPs) are cellular components recognised by the immune receptors of microglia, inducing or aggravating neuroinflammation in neurodegenerative disorders. Diverse evidence links mitochondrial dysfunction and inflammation mediated by mitochondrial-DAMPs (mtDAMPs) such as mitochondrial DNA, mitochondrial transcription factor A (TFAM) and cardiolipin, among others. Mitochondrial-derived vesicles (MDVs) are a subtype of EVs produced after mild damage to mitochondria and, upon fusion with multivesicular bodies are released as EVs to the extracellular space. MDVs are particularly enriched in mtDAMPs which can induce an immune response and the release of pro-inflammatory cytokines. Importantly, growing evidence supports the association between mitochondrial dysfunction, EV release and inflammation. Here, we describe the role of extracellular vesicles-associated mtDAMPS in physiological conditions and as neuroinflammation activators contributing to neurodegenerative disorders.

## 1. Introduction

Neuroinflammation can be defined as a primary defensive response of the brain against noxious stimuli that compromise the central nervous system (CNS) homeostasis [1]. An initial inflammatory response induces beneficial effects by promoting tissue repair and removing cellular debris. However, the beneficial outcomes can progress to deleterious consequences when chronic inflammation persists, inhibiting the cellular capacity to regenerate. The inflammatory response can endure due to endogenous factors, including genetic mutation or protein aggregation, or be triggered by environmental factors, including infection, trauma and drugs, and can lead to neurodegeneration [1]. Alterations in the microenvironment of the CNS, as in neurodegenerative disorders, can trigger an activation of the microglia and, thus, affect the development of neuronal networks and hasten the progress of the disease. The neuroinflammatory state linked to neurodegeneration induces the sustained release of pro-inflammatory molecules, which results in synaptic dysfunction, neuronal death and neurogenesis inhibition, leading to damage exacerbation [2]. In this sense, neuroinflammation tends to be a chronic process in which the persistence of nefarious stimuli is considered a driving force in the development of the neurodegenerative diseases such as Alzheimer’s disease (AD), Parkinson’s disease (PD), Huntington’s disease (HD) and amyotrophic lateral sclerosis (ALS), among others [3,4,5].

Inflammation comprises a complex crosstalk between the brain and infiltrating peripheral immune cells and is characterised by the production of (a) pro-inflammatory cytokines, namely, interleukin-1 beta (IL-1β), interleukin-6 (IL-6), interleukin-18 (IL-18) and tumour necrosis factor (TNF); (b) chemokines, including C-C motif chemokine ligand 1 (CCL1), CCL5 and C-X-C motif chemokine ligand 1 (CXCL1); (c) small-molecule messengers, such as prostaglandins and nitric oxide (NO); and (d) reactive oxygen species (ROS), by innate immune cells in the CNS [6] The chronic neuroinflammation is mediated by non-neuronal glial cells, including activated microglia and reactive astrocytes [7,8,9]. In this review, we describe the role of microglia cells in homeostasis and disease in the CNS and the implication of EVs-associated mitochondrial-derived components for microglia immune response.

## 2. The Role of Microglia in Brain Homeostasis and Neuroinflammation

Microglia cells are ubiquitously distributed in the brain, functioning as the main innate immune cells with a key role in the initial response to pathological insults. The main functions of microglia include (a) surveillance of the surrounding environment in the CNS; (b) acting as physiological housekeepers, migrating to injured sites, remodelling synapses and preserving myelin homeostasis; (c) protecting against nefarious stimulus, such as pathogen-associated molecular patterns (PAMPs) and damage-associated molecular patterns (DAMPs), which are described in detail below [10]. During non-physiological conditions, including recognition of foreign pathogens or other adverse stimuli, microglia are activated aiming to restore CNS homeostasis through the alteration in their secretory profile, morphology and phagocytic activity [11,12,13]. On the other hand, chronic inflammation can be triggered or maintained when a pro-inflammatory state in microglia is continuously activated, such as in response to primary neurodegeneration, axonal degeneration and processes linked to macrophages-mediated systemic inflammation [14,15,16].

In response to stimuli, activated microglia upregulate inducible nitric oxide synthase (iNOS) expression and the release of micromolar amounts of NO, and also cause the upregulation of NADPH oxidase (NOX) enzymes and subsequent intracellular ROS formation [17,18]. Under normal conditions, iNOS is not expressed in the brain. However, it can be detected in astrocytes and microglia in response to pro-inflammatory cytokines and pathogen components, including lipopolysaccharide (LPS), promoting neurodegeneration [19,20]. For several years, it was postulated that from a resting state, microglia can be polarised in two directions. The classical activation of microglia was known as M1 state, in which pro-inflammatory responses are predominant. An alternative microglial activation was known as M2, mainly responsible for resolution and repair, and represents an anti-inflammatory phenotype [21,22,23].

While M1 microglia polarisation was characterised by an increase in the expression of pro-inflammatory molecules, M2 microglial activation was further divided into four subtypes, M2a, M2b, M2c and M2d [24,25]. The M1 phenotype was characterised by several markers of cell surface, including CD11b, CD16, CD32 and CD86, and displays the pro-inflammatory response by expressing IL-1β, IL-6, IL-8 and TNF-α [26,27]. On the other hand, the M2 phenotype was characterised by other cell surface markers, including CD206, CD163 and arginase, and induces pro-inflammatory responses by expressing IL-10, IL-4, IL-13 and TGF-β [25,28].

The M1–M2 paradigm of microglial activation is artificial and only convenient to simplify the research in the field; currently, it is considered an oversimplified model that does not reflect the microglia complexity [29,30,31]. Accordingly, several microglia functions have been attributed considering their reaction through different phenotypes that are associated with distinct molecular signatures [32,33]. Moreover, microglia subtypes might display diverse intrinsic properties acquired during maturation or as a consequence of functional specialisations within the CNS. The coexistence at steady state of these subtypes allows them to undergo further modulation or phenotypic transformation in response to different *stimuli* [25,33,34]. Microglia might constitute a community of cells in which different members have several properties performing distinct physiological functions and responding differently to a particular stimulus [33,35]. The heterogeneity of microglia is related with several aspects, including temporospatial and gender-related differences regarding cellular origin, colonisation, abundancy, morphology, mobility (i.e., migration) and motility (extension and/or retraction of the processes), as well as gene expression, all of which ultimately reflect into diverse physiological and pathological functions [36]. The microglia morphology can be different in the presence of neuronal cell bodies, dendrites and axons, myelinated axons and blood vessels [32,35,37,38]. Under normal physiological conditions or after a stimulus (e.g., LPS), microglia show differences in self-renewal and turnover rates [35,38]. Using lipocortin 1 immunoreactivity as a brain-specific microglial marker, Savchenko et al. demonstrated that microglial density was higher in the forebrain, lower in the midbrain and lowest in the brainstem and cerebellum [39]. A more recent study, using time-lapse in vivo imaging, confirmed differences in the distribution and morphology between cerebellar and cortical microglia, with the latter having a higher microglial density and ramification compared to their counterparts [40]. However, the role of such microglia densities heterogeneity across distinct regions or even in the same region of the CNS remains unclear.

In most brain regions, microglia cells have a normal ramified morphology with extended branches, although significant variation in that morphology has been identified [41,42]. Vela et al. investigated the morphology and distribution of microglia in normal cerebellum of both young and adult mice through histochemical assays with nucleoside diphosphatase, a microglial marker. Importantly, this study demonstrated that microglia could have different sizes and ramification patterns not only within the same layer but also between different histological layers of the cerebellar cortex [42]. In summary, microglia, as a heterogeneous community of cells, can acquire different phenotypes which compromise important functional properties. These distinct subpopulations can coexist in the same brain structures and generate a non-uniform inflammatory response.

Strong evidence suggests that microglia are key regulators of inflammatory responses in neurodegenerative disorders, such as AD, PD, HD and ALS, among others. The distinctive neuroinflammatory state of these diseases promotes the continuous release of pro-inflammatory molecules that results in synaptic dysfunction, neuronal death and inhibition of neurogenesis, creating a vicious circle that exacerbates the damage [2]. Recently, a comprehensive single-cell RNA analysis of brain immune cells revealed disease-associated microglia (DAM) in the context of AD [43]. Several studies have since demonstrated the existence of neurodegenerative disease-associated phenotype of reactive microglia in AD, ALS and Frontotemporal Dementia (FTD) defined by a unique transcriptional and functional signature [43,44,45,46]. Most of these gene expression profiles were found in human genome-wide association studies (GWAS) associated with AD and other neurodegenerative disorders [19,43,47]. The DAM program is related to the expression of a subset of genes, such as TREM2 (triggering receptor expressed on myeloid cells 2), essential for its activation. In fact, studies using mouse models of neurodegenerative diseases deficient for TREM2 revealed that TREM2 signalling is essential for microglia to detect and respond to the “neurodegeneration cues”. TREM2 loss of function on microglial gene expression is associated with an increased risk for several neurodegenerative conditions, including AD, PD, ALS and FTD [48,49,50,51,52]. In *postmortem substantia nigra* and *putamen* brain tissue of sporadic PD, TNF-α, IL-1β, IL-6 and interferon-γ (IFN-γ) mRNAs levels are increased when compared to healthy controls [53,54]. In addition, in 6- hydroxy-dopamine (6-OHDA) treated mice, anti-inflammatory molecules, such as IL-10, were decreased, while pro-inflammatory cytokines, including IL-1β, IL-6, TNFα and IFN-ɣ, were upregulated [55]. TNF-α is highly expressed in dopaminergic neurons from PD patients and can exert a deleterious effect during an inflammatory response, such as triggering apoptotic cell death [56]. In rat hippocampal neurons, IL-1β treatment for 24 h (3 ng/mL) induces synaptic loss, depending on the simultaneous activation of both pre- and post-synaptic pathways [57]. A recent study using RNA sequencing analysis of microglia from mouse models for neurodegenerative disorders, namely, App^NL-G-F/NL-G-F^ (AD), rTg4510 (tauopathy) and SOD1^G93A^ (ALS) mice, demonstrated a reduction in the homeostatic microglial genes, whereas the DAM-associated genes were upregulated when compared to wild type (WT) [58]. In the same study, the gene expression of microglia-specific markers in *postmortem* brain tissue of early AD patients was also decreased, although the expression of DAM-related genes was not upregulated [58]. In the context of AD, another study demonstrated that the cerebral microcirculation participates in the inflammatory process as mediator. In fact, AD patients’ brain microvessels release higher levels of pro-inflammatory cytokines, such as TNF-α, IL-1β and IL-6, even under basal conditions, when compared to healthy control brains [59]. Importantly, anti-inflammatory cytokines, including IL-1 receptor antagonist, IL-4, IL-10 and IL-11, are also produced during the neuroinflammation process and could be part of a sophisticated mechanism to counteract excessive neuroinflammation [60,61,62]. In this sense, cerebrospinal fluid (CSF) from AD patients showed higher levels of pro-inflammatory, as well as anti-inflammatory cytokines, including eotaxin, IL-1ra, IL-4, IL-7, IL-8, IL-9, IL-10, IL-15, granulocyte colony-stimulating factor (GCSF), monocyte chemotactic protein 1 (MCP1), TNF-α and platelet-derived growth factor [63]. Importantly, a negative correlation between disease progression and the levels of several cytokines, such as IL-1β, IL-4, IL-6, IL-9, IL-17A, basic fibroblast growth factor, MCP1, IFN-γ and macrophage inflammatory proteins-1β was reported in the same study [63]. These effects might be dependent on the NLRP3 inflammasome activation since NLRP3 knock-out (KO) APP/PS1 mice for NLRP3 demonstrated decreased caspase-1 and IL-1β levels in the brain, enhanced amyloid-β (Aβ) clearance and were largely protected from spatial memory loss. Importantly, the microglia from those mice were switched to an anti-inflammatory phenotype (M2), promoting a decrease in Aβ levels [64]. The brain autopsy of multiple sclerosis patients evidenced a decrease in the microglial expression of Purinergic Receptor P2Y12 (P2RY12), a homeostatic microglial marker, as well as a higher susceptibility to a more pro-inflammatory phenotype, including the expression of phagocytic-related markers (macrosialin), antigen presentation markers (MHC class I and II molecules and T lymphocyte activation antigen CD86) and proteins involved in the production of ROS (cytochrome b-245 light chain) [65].

Although it seems clear that different microglia phenotypes are involved in physiological and pathological processes, the mechanisms involved in the regulation/activation of those different phenotypes remain unknown. It is likely that different phenotypes may be switched on by different triggers. In the context of neurodegenerative disorders where mitochondrial dysfunction is, alongside inflammation, one of the main hallmarks, it is conceivable that neuroinflammation could be mediated by released pro-inflammatory components by mitochondria. Thus, further studies are warranted to investigate not only the involvement of the different microglia phenotypes in physiological and pathological processes, such as neuroinflammation, but also to better understand how microglia respond to DAMPs through the expression of various immune receptors, including chemokine and pattern recognition receptors (PRRs), contributing to a gradual loss of neuronal function.

## 3. Damage-Associated Molecular Patterns Released from Mitochondria: Role in Neurodegenerative Disorders

For decades, the role attributed to the immune system was the recognition of “self” and “non-self”. In 1994, Polly Matzinger introduced the concept that the immune system’s “primary driving force is the need to detect and protect against danger” [66]. This new concept also introduced the idea that to work properly, the immune system does not stand alone but instead is connected to other cell types involving almost every bodily tissue [66]. In the late 1990s, it was discovered that the immune system can be activated through receptors called pattern recognition receptors (PRRs) [67]. PRRs are key components of the innate immune system able to stimulate phagocytosis and mediate inflammation [68]. Several families of PRRs are active within different domains of the cell [69]. The Toll-like receptor (TLR) family (TLR1 to TLR10) is mainly composed of type I transmembrane glycoproteins and can be located both at the cell surface and in intracellular membranes [70]. The nucleotide-binding oligomerisation domain (NOD)-like receptor (NLR) family resides within the cytoplasm and includes PRRs such as NODs, nucleotide-binding oligomerisation domain, Leucine rich Repeat and Pyrin domain-containing (NLRP) and IL-1B-converting enzyme (ICE)–protease-activating factor (IPAF) [67,71]. Other PRRs families include retinoic acid-inducible gene (RIG)-like receptors (RLRs), C-type lectin receptors (CLRs), formyl peptide receptors (FPRs) and scavenger receptors [69]. PAMPs are conserved structures on microbial pathogens while DAMPs, on the other hand, are molecules that can be released by stressed or dying cells and both can be sensed by PRRs, leading to the activation of the immune system [67]. Thus, the immune response induced by DAMPs occurs in the absence of infectious agents, a process known as “sterile inflammation” [72].

Several DAMPs have been identified over the years, with some of them originating from the extracellular matrix and others from different intracellular compartments including the cytosol, the plasma membrane and the nucleus [73]. Under physiological conditions, potential DAMPs presumably remain in the intracellular space and participate in the maintenance of cellular homeostasis [74]. However, under different active and passive mechanisms such as exocytosis of lysosomes/exosomes, apoptosis, necrosis, ferroptosis, pyroptosis and extracellular traps [75]. Ferroptosis is a newly described non-apoptotic form of cell death associated with lipid peroxidation and inflammation. Mitochondria seems to play a crucial role in ferroptosis through mitochondrial lipid metabolism, iron metabolism, energetic metabolism and other mitochondrial molecular regulatory processes [76]. Recently it was shown that ferroptotic cells are a rich source of DAMPs such as high mobility group box 1 (HMGB1) protein and adenosine triphosphate (ATP) [77]. Interestingly, the release of HMGB1 in ferroptosis is caused by nuclear membrane damage that anticipates cytoplasmic membrane rupture [78]. Damaged or dying cells release DAMPs to the extracellular space, and potentially induce a strong inflammatory response [79].

In fact, by interacting with PRRs and other immune cell receptors, DAMPs facilitate the mechanism of antigen presentation by dendritic cells, inducing the expression of pro-inflammatory molecules (such as the cytokines Il-1B and TNF-a) and NO in microglia from the CNS [80,81]. As such, DAMPs can cause the adverse activation of glial cells such as microglia [82]. Importantly, mitochondria have been recently presented as a main source of intracellular DAMPs [83,84,85]. Mitochondria play several important roles in the cell, including ATP production through oxidative phosphorylation (OXPHOS), calcium homeostasis, iron-sulphur clusters biogenesis, synthesis of heme and steroids, fatty acids catabolism, redox regulation of cellular signalling and regulation of apoptosis [86]. Given the endosymbiotic origin of mitochondria [87], mitochondrial molecules share many similarities with their bacterial counterparts [88]. As such, mitochondrial components such as mtDNA, cardiolipin, cytochrome c (CytC), TFAM and N-formyl peptides can be recognised by the immune system through the interaction with PRRs, triggering an innate immune mechanism [89]. Molecules released by mitochondria which trigger immune responses are thus considered as mitochondrial DAMPs (mtDAMPs).

Neurodegenerative diseases often share similar pathological processes, such as mitochondrial dysfunction, excitotoxicity, oxidative stress and inflammation. Although the specific mechanisms underlying each neurodegenerative disorder are unknown, it is consensual that mitochondrial abnormalities are critical to trigger cytotoxic events that ultimately lead to neuronal death [90]. Moreover, there is a growing recognition that neuroinflammation does not act independently and may be intricately linked to mitochondrial dysfunction. Importantly, in most neurodegenerative disorders, the inflammation is not driven by a pathogen, suggesting that neuroinflammation might be triggered by autologous molecules. Excessive mitochondrial damage associated to impaired mitophagy, commonly observed in neurodegenerative disorders [91], can lead to the release of mtDAMPs into the cytoplasm and extracellular environment, thus enhancing the neuroinflammatory process [92]. Moreover, sustained mitochondrial dysfunction leads to cell death, which is aggravated in the CNS due to the high metabolic demand and the limited capacity of neuronal regeneration [84]. Thus, a deeper understanding of the crosstalk between mtDAMPs and neuroinflammation, probably involved in the neurodegeneration process, may contribute to the identification of new therapeutic targets.

### 3.1. Mitochondrial DNA

MtDNA is a small circular molecule of double-stranded DNA (dsDNA) that encodes 13 respiratory chain proteins as well as the transfer and ribosomal RNAs needed for their translation in the mitochondrial matrix [93]. MtDNA also encodes mitochondria-derived peptides including humanin, mitochondrial ORF of the twelve S-c (MOTS-c), and small humanin-like peptides (SHLPs 1–6), which are regulators of metabolism [94,95]. Besides being hypomethylated, when compared to nuclear DNA [96], mtDNA is more prone to oxidative damage given its lack of structural histones and proximity to sources of ROS from the electron transport system [97]. Nevertheless, recent reports highlight the existence of mtDNA maintenance proteins that are shared with the nucleus. PrimPol is a primase–polymerase involved in mtDNA damage response [98]. The first evidence of mtDNA as DAMPs was published in 2004 by Collins et al., who induced inflammatory arthritis through the injection of mtDNA in the knees of 6- to 8-week-old mice [99]. Since then, several studies have shown that mtDNA can activate several PRRs. During severe stress, mtDNA is released from the mitochondria, activating the neuroinflammatory response by triggering interferon (IFN) genes, TLR9 and the NLRP3 inflammasome [100]. MtDNA is detected by cGAS, a dsDNA sensor that generates cGAMP that further binds directly to the stimulator of interferon genes (STING), activating the protein kinase tank-binding kinase 1 (TBK1). The activation of TBK1 leads to interferon regulatory factor 3 (IRF3) phosphorylation, translocation into the nucleus and consequent induction of various IFN-stimulated genes [101,102,103]. Moreover, TBK1 also activates the NF-κB signalling pathway through phosphorylation, thus increasing the expression of IL-6 and TNF-α. Furthermore, through the binding to TLR9, mainly present in the membrane of the endosome, mtDNA can trigger a downstream signalling cascade, which activates MAPK and the nuclear transcription factor NF-κB. MtDNA can also activate the NLRP3 inflammasome, and through the caspase-1 activation, promote the expression of IL-18 and IL-1β [84,104]. MtDNA internalisation can possibly be attributed to mechanisms such as endocytosis, phagocytosis and/or transmembrane diffusion; however, further clarification is required to properly explain the activation of intracellular PRR signalling factors under such circumstances [97].

In a cellular model of HD (differentiated Q111 cells vs Q7), the mtDNA release is significantly increased and accompanied by the activation of the cGAS/STING/IRF3 pathway and, consequently, inflammation [105], evidencing the role of mtDNA as a DAMPs in HD. In ALS patients, induced pluripotent stem cells (iPSC)-derived motor neurons, TDP-43 mutant mice and NSC-34 cells expressing a doxycycline-induced ALS TDP-43 gene, inflammation is activated through the cGAS/STING pathways after TDP-43 enters mitochondria and triggers mtDNA fragments release via the permeability transition pore [106], identifying mtDNA as a determinant trigger in TDP-43-associated pathology. In PD patients, the levels of circulating mtDNA in the CSF are significantly reduced [107], and are associated with cognitive impairment and also influenced by treatment [108]. The reduction of circulating mtDNA in the CSF may be explained by the PD-associated decrease in mtDNA transcription factors TFAM and TFB2M and mitochondrial depletion [109]. In 2019, Picca et al. published the rationale, design and methodology of the EXosomes in PArkiNson Disease (EXPAND) study [110], which aims to characterise the cargo of small circulating EVs in PD patients. However, the study did not publish the results regarding the release of mtDNA into EVs. Moreover, there are no data regarding the levels of extracellular mtDNA in PD brain. Importantly, mitochondrial dysfunction caused by respiratory chain complex deficiencies associated with mtDNA depletion and/or mutation, such as the one observed in PD [109,111], can be a potential source of mtDAMPs. Furthermore, both Prkn-/- and Pink1-/- mice, which accumulate mtDNA mutations and, consequently, mitochondrial dysfunction, present a strong inflammatory phenotype depending on STING-mediated type I interferon response [112]. Levels of circulating mtDNA are also decreased in the CSF of asymptomatic patients at risk of AD and symptomatic AD patients, which can be associated with a decrease in the number of mtDNA copies in neurons [113]. To date, there are no data regarding the release of mtDNA into the extracellular space and its impact on AD-associated neuroinflammation. However, considering the existing data, and similar to our PD hypothesis, it is likely that the mitochondrial dysfunction observed in AD is accompanied by the release of mtDNA, leading to the amplification of the neuroinflammatory process observed along the disease.

Thus, mtDNA appears as potential relevant mtDAMPs involved in the neuroinflammation associated with neurodegenerative disorders. However, further studies are needed to confirm its implication.

### 3.2. ATP

ATP is the most direct source of energy in organisms. In the nervous system, ATP can be stored in the presynaptic vesicles and granules of neurons and glial cells. The release of ATP into the extracellular space during chronic neurodegenerative disorders is believed to be due to a loss of integrity of the plasma membrane of neuronal cells [114]. Importantly, ATP can act as a signalling molecule in immune and inflammatory response [115]. Thus, when ATP is released into the extracellular space, it activates the microglial purinergic receptors (PRs), ionotropic P2X and metabotropic P2Y, mediating the release of pro-inflammatory cytokines that can promote neurodegeneration [116]. The P2X7 receptor, mainly localised in microglia, is an ATP-gated ion channel which also allows the penetration of larger organic molecules and, when activated, indirectly leads to the activation of NLRP3 inflammasomes, which further triggers the activation of caspase-1 signalling cascade [117,118].

In AD, the TLR4/NLRP3 pathway plays a critical role in Aβ-induced neuroinflammation and probably is related to the increase in extracellular ATP. Aβ1-42 activates and upregulates the expression of NLRP3 inflammasome in BV-2 microglia, as indicated by increased activation of caspase-1 and secretion of IL-1β [119]. Importantly, this activation might be associated with the release of ATP through pores present in the cellular membrane and formed by Aβ peptides [120]. In hippocampal neurons, this extracellular ATP further leads to the increase in intracellular calcium through P2XR activation [120], increasing excitotoxicity.

In PD, the inhibition of P2X7 receptor in 6-OHDA-induced PD-like symptoms in rats improves both animal behaviour as well as microgliosis and astrogliosis [121,122], evidencing the probable role of extracellular ATP in disease etiopathogenesis. Importantly, extracellular ATP induced a significant increase in intracellular α-synuclein protein levels associated with the increase in Ca^2+^ influx via the activation of P2X1 receptor in PD neuronal cells [123]. Moreover, extracellular α-synuclein is recognised by P2X7 receptors in microglial cells and induces oxidative stress in a process dependent on the PI3K/AKT activation [124]. In neurons, extracellular α-synuclein induces the activation of the P2X7 receptor, which leads to the recruitment of pannexin 1, an ATP-permeable channel, further responsible for ATP release [125]. Moreover, α-synuclein decreases the activity of ecto-ATPase responsible for ATP degradation [125]. Thus, the release of ATP increases the accumulation of α-synuclein, which further increases cellular dysfunction and exacerbates the release of DAMPs and mtDAMPs, such as ATP, forming a self-perpetuating inflammatory response.

Mouse and cellular HD models show significantly increased levels of P2X7 receptor and altered P2X7-mediated calcium permeability in somata and terminals of neurons [126]. The same upregulation has been observed in HD *postmortem* brains [127]. In cellular models, this increase is accompanied by a higher susceptibility to apoptosis triggered by P2X7 receptor stimulation [126]. Although there is no evidence of changes in the levels of extracellular ATP in the brain of HD patients and mouse models [128], changes in P2X7 receptor might be associated with an excessive response to mtDAMPs in HD.

On the other hand, increased ATP levels in ALS patients CSF were measured [129]. Importantly, Apolloni et al. demonstrated that the antagonism of P2X7 receptor in a SOD1-G93A mouse model of ALS ameliorates the early symptomatic phase of the disease by reducing microglia-related pro-inflammatory markers and autophagy [130]. In fact, the activation of P2X7 receptor in SOD1-G93A mouse leads to the increase in autophagic flux in microglia [131] and increased oxidative stress [132] concomitantly with the enhancement of pro-inflammatory action of microglia [133].

Altogether, these data evidence the role of ATP as a mtDAMP in several neurodegenerative disorders and the potential of targeting P2X7 receptors to at least delay disease progression through the modulation of inflammatory response.

### 3.3. Cytochrome C

CytC, a mitochondrial protein essential for the proper functioning of the electron transport chain (ETC) and a potent ROS scavenger [134], may also be considered a DAMP under cellular stress. In healthy cells, CytC is in the mitochondrial intermembrane/interscristae spaces, where it functions as an electron shuttle in the mitochondrial respiratory chain and interacts with cardiolipin. Under stressful conditions, CytC can function as a peroxidase, eventually resulting in the oxidation of cardiolipin [135], a mechanism that facilitates CytC release into the cytosol. Once in the cytosol, CytC facilitates the assembly of apoptosomes, mediating the activation of the caspase cascade that further leads to apoptosis [136].

Evidence shows that CytC can be also released into the extracellular space by damaged or dying cells and can be used as a marker of cell death by apoptosis in vitro and in vivo [137]. Several studies reported the immunomodulatory properties of extracellular CytC in the periphery. Thus, the intra-articular injection of CytC in mice leads to chronic inflammation through the accumulation of neutrophils and macrophages [138]. This activation seems to occur through the induction of nuclear factor (NF)-κB activation and the release of inflammatory cytokines and chemokines such as interleukin (IL)-6, tumour necrosis factor (TNF)-α and monocyte chemoattractant protein (MCP)-1, as shown in cultured mouse splenocytes [138]. In primary neuronal cultures, the presence of CytC into the extracellular space led to increased staurosporine-induced apoptosis [139]. Moreover, in primary human astrocytes, extracellular CytC increases the secretion of IL-1b and IL-8 also through the binding and activation of TLR4 [140]. Importantly, the extracellular release of CytC via EVs was previously described as being increased in HD human iPSC compared to controls alongside increased mitochondrial dysfunction [141]. To our knowledge, there are no studies assessing the extracellular release of CytC in the context of AD, PD or ALS. However, there is evidence of the release of CytC from mitochondria into the cytoplasm in each of these neurodegenerative disorders [139,140]. Thus, we hypothesise that due to cellular loss of membrane integrity, some of the CytC pool release by mitochondria to the cytoplasm further leaks to the extracellular matrix, participating in the activation of inflammatory response, serving thus as a mtDAMP.

### 3.4. TFAM

TFAM is a member of the high-mobility group box (HMGB) protein family. TFAM compacts mtDNA into nucleoids and is essential to ensure mtDNA homeostasis, regulating DNA transcription and, consequently, mtDNA copy number. TFAM protein levels regulate mtDNA transcription by exposing a specific binding site for mitochondrial promoters, thus allowing their activation. When mitochondria are damaged, TFAM, located at the matrix, can be released to the extracellular space and activate inflammation [142]. TFAM has been implicated in the inflammation of the CNS in neurodegenerative diseases [84]. In fact, Schindler et al. demonstrated that extracellular TFAM injected into the cisterna magna of rats induces inflammation and cytotoxicity through the upregulation of monocyte chemotactic factor 1 (MCP-1), NF-κB, IL-6 and TNF-a in the hippocampus, and MCP-1, IL-1B and TNF-a in the cortex [143]. The release of TNF-α from reactive dendritic cells under TFAM exposure occurs through the activation of TLR4 and the receptor for advanced glycation end-products (RAGE) [143,144].

In AD, RAGE can be activated by direct binding of Aβ [145]. Moreover, AD brains present an increased number of RAGE-immunoreactive microglia and seem to be more susceptible to Aβ-induced microglia activation than control [146]. Therefore, the TFAM-RAGE pathway could be potentially involved in disease progression mechanisms. Similar observations can be made in other neurodegenerative disorders. For example, in several models of PD, the activation of RAGE has been described [147], mainly through the extracellular release of the HMGB1 [148]. The same mechanism is also activated in ALS [149] and HD neurons [150], and HD astrocytes [151] evidence higher levels of RAGE. As for AD, we cannot exclude the possible release of TFAM by damaged cells into the extracellular space and the consequent activation of RAGE and the triggering of inflammation observed in PD and/or ALS.

### 3.5. Cardiolipin

Cardiolipin is a phospholipid primarily found within the inner mitochondrial membrane (IMM), constituting about 20% of the total lipid composition. Cardiolipin plays an essential role in several reactions and processes including the structural and functional integrity of several IMM proteins, such as enzyme complexes of the ETC, and for their organisation into supercomplexes. Furthermore, it also regulates mitochondrial morphology, stability and dynamics [152]. Besides its extensively studied role as an essential regulator of metabolic processes, cardiolipin is reportedly involved with the regulation of mitophagy, an autophagic mechanism destined for the elimination of dysfunctional mitochondria [84]. However, when mitochondria are damaged, cardiolipin may be exposed to the mitochondrial surface, exerting pro-mitophagy [153] or pro-apoptotic signals [154]. Importantly, cardiolipin regulates the microglial function through increasing microglial phagocytosis [155] in a TLR4-dependent manner [156]. Moreover, there is evidence that cardiolipin directly binds and activates NLRP3, activating the inflammasome, resulting in a neuroinflammatory response [157,158].

The modulation of NLRP3-mediated inflammasome activation has been presented as a potential therapeutic approach in AD [159,160], PD [161], HD [162] and ALS [163]. Although the relationship between NLRP3 and extracellular cardiolipin has not been assessed in these neurodegenerative diseases, it seems legitimate to believe in the involvement of cardiolipin as a mtDAMP in neurodegeneration progression. Extracellular cardiolipin significantly upregulated the phagocytic activity of primary microglia isolated from C57BL/6 mouse brains [155]. Moreover, extracellular cardiolipin reduces the secretion of the pro-inflammatory, and potentially cytotoxic, TNF-α by Aβ- and α-synuclein-stimulated primary murine microglia [156]. Thus, following its release from cells, cardiolipin may act as a DAMP by regulating cytokine release and phagocytic activity of the surrounding immune cells in the CNS, participating in the neuroinflammation process.

Altogether, these observations support the hypothesis that mtDAMPs are involved in the process of neuroinflammation observed in neurodegenerative disorders. Going one step forward, it is sensible to think that mitochondrial dysfunction and neuroinflammation work together to induce neurodegeneration.

## 4. Mitochondrial DAMPs in Extracellular Vesicles as Inflammasome Activators

EVs are lipid membrane-enclosed, nanosized particles secreted into the extracellular space by virtually all types of cells (Figure 1). EVs cargo is determined by their donor cell and includes RNA, DNA, lipid, and proteins, among others cell-specific molecules (Figure 1) [164]. Once internalised, EV-associated components can modulate intracellular pathways and modify functions in the recipient cells. EV subtypes can be classified, based on the size, into small EVs and medium/large EVs, with ranges defined <200 nm for small and >200 nm for large [165]. EVs of endosomal origin are a subtype of EVs known as “exosomes” (30–150 nm) that are released upon the fusion of late endosomes/multivesicular bodies (MVBs) to the plasma membrane [166]. Exocarta (www.exocarta.org/; accessed on 1 June 2022), EVpedia (http://evpedia.info/; accessed on 1 June 2022), EVmiRNA (http://bioinfo.life.hust.edu.cn/EVmiRNA; accessed on 1 June 2022), Vesiclepedia (www.microvesicles.org/; accessed on 1 June 2022), ExoRbase (www.exoRBase.org/; accessed on 1 June 2022) and EV-TRACK (http://evtrack.org; accessed on 1 June 2022) are EV-related databases which centralise the knowledge on EVs cargo.

Several studies have shown that mitochondrial components are secreted by cells under physiological and pathological conditions [92,167,168,169]. Moreover, numerous mitochondrial proteins have been identified inside EVs, such as TFAM, TOM20, CytC, PDH, mtHSP70 (mitochondrial matrix), NDUFA9 (mitochondrial Complex I subunit), SDHA (mitochondrial Complex II), UQCRC2 (mitochondrial Complex III), OPA1, ATP5A and VDAC1 [92,170,171,172]. The discovery of mitochondrial components as part of EVs cargo has identified several biological functions linked to extracellular mitochondria elements, including restoration of mitochondrial function and metabolic rescue [168], mitochondrial degradation [173] and inflammation [89,170].

Mitochondria seems to play a role on the NLRP3 inflammasome [174]. Thus, the improvement of mitochondrial viability through the modulation of PGC-1α attenuated the NLRP3 inflammasome [175]. Mitochondrial ETC function has also been linked to NLRP3 activation through mechanisms dependent on ROS generation [7,176] or ATP-mediated via a ROS-independent mechanism [177] that requires intact mitochondria [178]. Moreover, damaged mitochondria can stimulate the release of mtDAMPs cargo within EVs [110]. Additionally, Todkar et al. proposed that to prevent the secretion of oxidised proteins and consequent activation of the immune system, cells can selectively target ROS-induced mitochondrial proteins containing EVs to lysosomes for degradation [170]. However, our lab recently showed that an impairment in the mitochondrial–lysosomal axis can promote the release of mitochondrial DAMPs in EVs [92]. Mitochondrial components can be secreted through the endocytic pathway and later be released into the extracellular space as EVs [179]. This mechanism can enable the uptake of mitochondrial components by recipient cells to further activate an innate immune response [179]. On the other hand, the activation of the inflammasome causes the enhancement of EVs secretion, probably as a strategy to mitigate inflammation and prevent cell damage. Thus, a recent study showed that inflammasome-activated macrophages release interferon β (IFNβ)-containing EVs that induce an interferon signature in bystander cells and result in moderating the NLRP3 inflammasome responses [180]. Moreover, treatment with bone marrow-derived mesenchymal stem cell (MSC)-derived exosomes prevent the H_2_O_2_-induced inflammatory mediators and NLRP3 inflammasome activation [181]. In this study, 10.7% of EV proteins were mitochondria-related, suggesting that the protective effects of EVs could be due to the supply of “healthy” mitochondrial proteins to damaged mitochondria, therefore attenuating the mitochondrial dysfunction and ROS production [181]. In this sense, immune modulatory activity of MSC has been described in numerous studies pointing out that their positive effects are partially mediated by the various components of MSC derived-EVs [182].

Thus, EVs’ mitochondrial cargo effects can be ambivalent either by attenuating or enhancing the inflammatory response. The cellular origin of EVs is probably the major contributor for this discrepancy, along with the conditions that induce cell release of EVs and determine the molecular contents.

## 5. Mitochondrial-Derived Vesicles and Mitovesicles

Mitochondrial-derived vesicles (MDVs) were first described in 2008 as vesicles released by mitochondria carrying selective cargo [183]. The generation of MDVs has been mainly associated with increased levels of intracellular ROS [173] and is suggested to be part of the mitochondrial quality control complementary to mitophagy. In vitro work has described that MDVs cargo is mainly dependent on the nature of mitochondrial stress involved. For instance, the exposure to ROS generated outside mitochondria leads to vesicles enriched for VDAC, but lacking complex III [184]. Moreover, although in basal conditions, MDVs already carry oxidised proteins, probably as an ongoing maintenance process, under acute stress, MDVs became enriched in oxidised proteins [184].

MDV trafficking precedes the degradation of mitochondrial proteins through mitophagy [185] and can be an adaptive mechanism to autophagy deficiencies [186]. TOM20-positive MDVS were described as containing mitochondrial extracellular membrane proteins [170]. Importantly, under oxidative stress conditions, the incorporation of oxidised mitochondrial components into TOM20-positive MDVs primarily targeted for lysosomal degradation is enhanced [187,188]. This process can be orchestrated in a PINK1/Parkin-dependent manner [170,185] and targeted to Tollip (Toll-interacting protein)-positive endosomes [189], facilitating the MDV trafficking to the lysosomes [190]. Another alternative secretory pathway is the fusion of MDVs with MVBs [173], which generates EVs further released to the extracellular space [191]. The routing of mitochondrial inner-membrane/matrix to EVs is OPA1- and Snx9-dependent [170], contrary to TOM20-positive MDV formation [170]. Moreover, inner-membrane/matrix-containing MDVs are inhibited by oxidative stress [170]. MDVs positive for phosphatidic acid require MIRO1/2 and DRP1 for their formation [192]. Thus, although MDV generation was initially described to be a DRP1-independent mechanism [183], data suggest that MDV biogenesis has different mechanisms that dictate the cargo selectivity and targets. Beside the lysosomal transport, another subtype of MDVs, containing MAPL protein but lacking TOM20, transports cargo from the mitochondria to the peroxisomes [183], being this mechanism involved in the maintenance of peroxisome morphology [193]. The role of MAPL may be in regulating both peroxisomal and mitochondrial morphology, similar to the other proteins involved in fission. Moreover, the crosstalk between peroxisomes and mitochondria suggests that MAPL-positive MDVs are involved in this communication as transporters of metabolites, lipids or proteins to a peroxisome subpopulation [191,194].

The encapsulation of mitochondrial material/DAMPS in EVs has been reported as a coping mechanism to avoid the release of free mtDAMPs involved in toxic responses in other organelles or cells [170]. Although some reports have shown that the transport of whole healthy mitochondria through EVs can have a beneficial effect on failing chondrocytes [195] and in mtDNA-deficient L929 Rho0 cells and mononuclear phagocytes in a multiple sclerosis animal model [168], others have reported the induction of a pro-inflammatory response in endothelial cells [196].

MDVs were reported to be involved in a PD-associated mechanism in which VPS35, a protein of the membrane protein-recycling retromer complex that is mutated in PD, increasingly interacts with dynamin-like protein 1 and enhances its lysosomal degradation through the MDVs pathway [197]. Additionally, the analysis of small EVs cargo isolated from serum of 70+ years PD patients identified mitochondrial proteins such as adenosine triphosphate 5A, NADH:ubiquinone oxidoreductase subunit S3 and succinate dehydrogenase complex iron sulphur subunit B in lower levels associated with an increased release of small EVs when compared to controls [198]. Another study has shown that EVs isolated from astrocytoma cells treated with Aß42 oligomers had higher levels of mitochondrial proteins (TOM20 and OPA1) when compared to non-treated cells, showcasing the possible use of mitochondrial proteins within EVs as AD biomarkers [199]. Moreover, neuron-derived EVs isolated from AD patients’ plasma carried lower levels of mitochondrial complexes I, III, IV and V, as well as SOD1, and decreased catalytic activity of complex IV and ATP synthase compared to controls [200]. A proteomic analysis of EVs content from AD brains (grey matter tissue from the frontal cortex) showed that 148 proteins were exclusive to the AD samples and were linked to mitochondrial function by Gene Ontology analysis [201]. In HD, TFAM was detected in HD human fibroblast-derived EVs [92] and CytC in human iPSC-derived neural stem cell EVs [141]. Besides mitochondrial proteins, mtDNA and mtDNA-RNA were identified in EVs [92,159,170,171,172,199]. Although EV-associated mtDNA was shown to rescue oxidative phosphorylation deficits in breast cancer cells [171], the pro-inflammatory action of EV-related mtDNA has also been described [202]. In fact, the levels of platelet-derived EVs containing mitochondria were positively correlated with the levels of mtDNA present in these EVs and adverse reactions related to platelet transfusion [202].

In 2021, a new subset of mitochondrial EVs was reported and the term “mitovesicles” was employed to describe double-membraned EVs carrying mitochondrial proteins [203]. The authors applied a high-resolution gradient separation gradient to isolate brain-derived EVs from WT mice according to their density. Eight fractions were obtained, with each one containing different sized EVs carrying different cargo. The denser fraction contained mainly double-membraned EVs lacking the commonly used markers Alix, TSG101 and CD63 and enriched in proteins from the mitochondrial outer (VDAC)/inner (COX-IV) membranes and matrix (PDH-E1). Cardiolipin was also highly present in this fraction, and a total of 279 mitochondrial proteins were detected by mass spectrometry, including mitochondrial proteins expressed by astrocytes and neurons. In contrast to what was seen before in MDVs, TOM20 and DRP1 were not detected in mitovesicles as they were in exosomal fractions. Additionally, mitovesicles showed enzymatic activity and capacity for ATP production. Down syndrome mice brain-derived mitovesicles showed higher mitochondrial protein content and mtDNA levels than controls [203]. These data confirmed the relation between mitochondrial alterations in neurodegeneration and mitochondrial cargo within mitochondria-specific vesicles, here mentioned as mitovesicles. Currently, to our knowledge, there are still no new studies regarding the classification and analysis of mitovesicles and their differences relative to MDVs. They seem to have different mitochondrial cargo and express different markers than MDVs, which adds to the need of continuing to explore these mitochondrial-specific vesicles and their link to mitochondrial dysfunction described in neurodegenerative diseases.

## 6. Concluding Remarks

Metabolic and mitochondrial dysfunction occur early in many neurodegenerative diseases. Dysfunctional mitochondria are a source of DAMPs which can trigger inflammation, a potent driver in the pathogenesis of brain diseases. Importantly, mtDAMP could facilitate neuroinflammation when mitochondrial dysfunction overcomes the neurons’ capability to adequately perform mitochondrial degradation through autophagy. In this sense, autophagy can directly control inflammation by mediating a negative feedback mechanism. In addition, the normal secretion of mitochondrial DNA and proteins within EVs by functional cells is significantly increased in the presence of mitochondrial stress, probably in an attempt to modulate uncontrolled mtDAMPs release. However, it is likely that EVs, which have been implicated in the induction of inflammasome, exert under stressful conditions a pro-inflammatory action mediated via the EV-associated mtDAMPs to the surrounding cells. Thus, mitochondrial dyshomeostasis and neuroinflammation may synergistically trigger a vicious cycle resulting in neuronal death (Figure 2). The mechanisms linking both processes have not yet been elucidated, thus opening new avenues for interference with microglia-driven cerebral inflammation and delaying the process onset.

## Figures and Tables

**Figure 1 cells-11-02364-f001:**
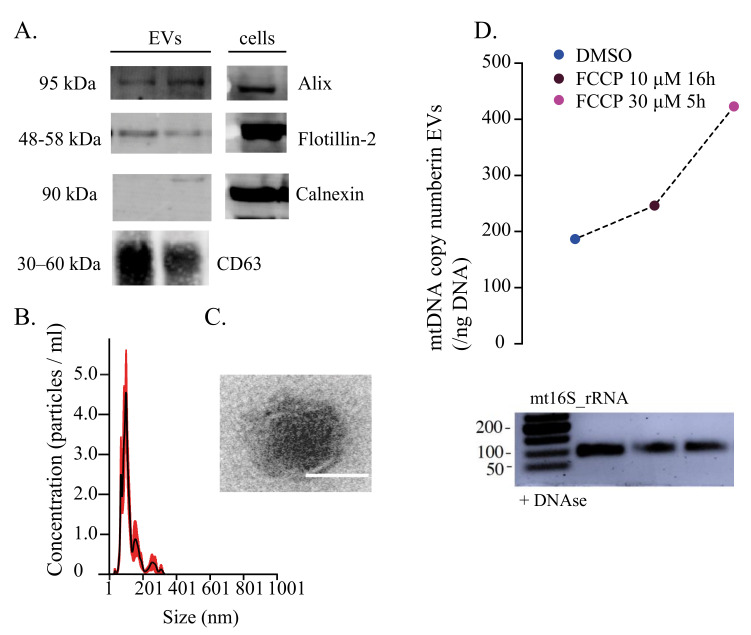
Characterisation of human extracellular vesicles carrying mitochondrial DNA. (**A**) Western blots of human fibroblasts-derived EVs with traditional protein markers Alix, Flotillin-2 and CD63. Calnexin was used as a negative control for cell contamination. (**B**) Nanosight tracking analysis of EV concentration (particles/mL) and size (nm). (**C**) TEM image of the EV structure with visible lipidic bilayer. Scale: 100 nm. (**D**) Quantification of mtDNA copy number (/ng DNA) isolated from DNAse-treated EVs. Prior to EVs’ purification, cells were treated with 10 μM FCCP for 16 h and 30 μM FCCP for 5 h. DMSO was used as a control. PCR gel electrophoresis of mt16S isolated from DNAse-treated EVs. EVs: extracellular vesicles, TEM: transmission electron microscopy. Adapted from https://doi.org/10.1101/2022.02.13.480262; accessed on 3 June 2022.

**Figure 2 cells-11-02364-f002:**
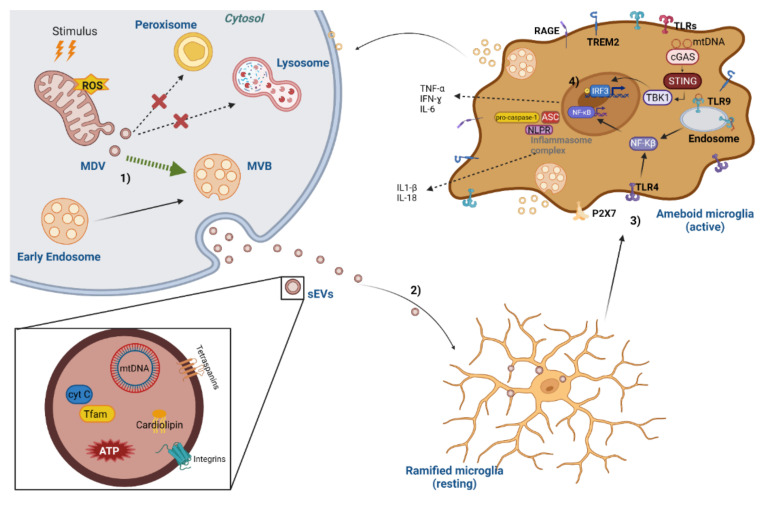
Molecular mechanisms of EV-associated DAMPs-mediated neuroinflammation. (1) After an oxidative stimulus, increased levels of reactive oxygen species (ROS) lead to generation of mitochondrial-derived vesicles (MDVs). The process of MDV production can result in delivering oxidised proteins to peroxisome/lysosome degradation or routing them to the multivesicular bodies (MVBs), which will be released as extracellular vesicles (EVs). Several mitochondrial components can be released within MDVs, including mtDNA, cardiolipin, cytochrome C, mitochondrial transcription factor A (TFAM) and N-formyl peptides, which are considered as mitochondrial damage-associated molecular patterns (DAMPs). (2) Mitochondrial DAMPs in EVs can initiate a microglia pro-inflammatory immune response by inducing a conformational change from a normal ramified morphology (resting) to an ameboid form (activated). (3) Mitochondrial DAMPs interact with pattern recognition receptors that include the Toll-like receptor (TLR) family (TLR1 to TLR10). MtDNA can activate a neuroinflammatory response by triggering interferon genes, TLR9 (membrane of the endosome) and the NLRP3 inflammasome. MtDNA can elicit a downstream signalling cascade which activates MAPK and the nuclear transcription factor NF-κB. MtDNA can also activate the NLRP3 inflammasome and through the caspase-1 activation promote the expression of IL-18 and IL-1β. (4) MtDNA is also detected by cGAS, that further binds to STING activating TBK1. The activation of TBK1 leads to IRF3 phosphorylation, translocation into the nucleus and consequent induction of various IFN-stimulated genes. TBK1 also activates the NF-κB signalling pathway, thus increasing the expression of IL-6 and TNF-α, contributing to chronic neuroinflammation and accelerating neuron degeneration. Abbreviations: ATP—adenosine triphosphate; cyt C—cytochrome C; MBVs—multivesicular bodies; MDV—mitochondrial-derived vesicle; mtDNA—mitochondrial deoxynucleic acid; NLPR—NOD-like receptor; RAGE—receptor for advanced glycation end-products; ROS—reactive oxygen species; TFAM—mitochondrial transcription factor A; TLRs—Toll-like receptors; TREM2—triggering receptor expressed on myeloid cells 2; STING—stimulator of interferon genes; TBK1—protein kinase tank-binding kinase 1; IRF3—interferon regulatory factor 3. Created with BioRender.com; accessed on 12 June 2022.

## Data Availability

Not applicable.

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
