# Peer review of "Mitochondrial Damage-Associated Molecular Patterns Content in Extracellular Vesicles Promotes Early Inflammation in Neurodegenerative Disorders"

_cells, 2022, doi:10.3390/cells11152364_

Round 1

Reviewer 1 Report

Dear authors, 

I am pleased to read the review article written about mtDAMPS and neurodegenerative disorders. In my understanding, this is a good work that bing together the latest findings about the theme; which is a hotspot of neuroinflammation nowadays. 

The review text is well structured by presenting fundamental concepts to the reader to understand the deeply content that comes after the discussion. I really appreciate this way of writing. 

I have found that reference 186 may be not correct by comparing the information in the reference with the information in the text, please double-check the references. 

Besides that, I have found no major issues. 

Figure 2 at the end of the text is also very helpful to memorize graphic information from the review literature. 

Best regards, 

Author Response

I am pleased to read the review article written about mtDAMPS and neurodegenerative disorders. In my understanding, this is a good work that bring together the latest findings about the theme; which is a hotspot of neuroinflammation nowadays.

The review text is well structured by presenting fundamental concepts to the reader to understand the deeply content that comes after the discussion. I really appreciate this way of writing. 

1) I have found that reference 186 may be not correct by comparing the information in the reference with the information in the text, please double-check the references. Besides that, I have found no major issues. Figure 2 at the end of the text is also very helpful to memorize graphic information from the review literature. 

Answer: The authors are grateful for the reviewer’ comment and apologies for the mistake. Indeed, reference 186 (reference 189 of the revised manuscript) was not correct and was changed (Page 12 of the revised manuscript). As recommended by the reviewer, the authors double-checked the references throughout the manuscript.

Reviewer 2 Report

Deus et alreview .'s demonstrated the involvement of mitochondrial DAMP and its relationship with extracellular vesicles in neurodegenerative inflammation. The manuscript is well written and has clinical relevance not only in neurological disorders but also in other conditions. There are only minor issues.

1. The authors investigated every aspect of mitochondrial DAMP that has been linked to neurodegenerative diseases. "Ferroptosis," which is mostly regulated by mitochondria, is one of the developing fields in cell death. It would be fascinating if the authors could describe the relationship between mitochondrial DAMPs and ferroptosis.

2. The reference to line 411 on page 8 is missing.

Author Response

Deus et al. review's demonstrated the involvement of mitochondrial DAMP and its relationship with extracellular vesicles in neurodegenerative inflammation. The manuscript is well written and has clinical relevance not only in neurological disorders but also in other conditions. There are only minor issues.

1) The authors investigated every aspect of mitochondrial DAMP that has been linked to neurodegenerative diseases. "Ferroptosis," which is mostly regulated by mitochondria, is one of the developing fields in cell death. It would be fascinating if the authors could describe the relationship between mitochondrial DAMPs and ferroptosis.

Answer: The authors thank the reviewer for the suggestion and a paragraph describing the involvement of ferroptosis, mitochondria and DAMPs was added. (Page 5, lines 235 to 243 of the revised manuscript).

2) The reference to line 411 on page 8 is missing. 

Answer: The authors are thankful for pointing out this error. The references (138 and 139) were added accordingly (Page 8, line 421 of revised manuscript).